# Correlation between relative afferent pupillary defect and visual field defects on Humphrey automated perimetry: A cross-sectional clinical trial

**Juthamat Witthayaweerasak**[ID]\*, **Pemika Lertjittham**[ID]°, **Nipat Aui-aree**°

Department of Ophthalmology, Faculty of Medicine, Prince of Songkla University, Hatyai, Songkhla, Thailand

° These authors contributed equally to this work.

\* juthamat.w@psu.ac.th

**Data Availability Statement:** All relevant data are within the manuscript and its Supporting Information files.

## Abstract

### Purpose

To evaluate the correlations between relative afferent pupillary defect (RAPD) magnitude, assessed using the clinical plus scale and neutral density filters, and visual field parameters in patients with unilateral or asymmetrical bilateral optic neuropathy or retinopathy.

### Methods

Fifty-two patients with RAPD, graded by the swinging flashlight test and neutral density filters, were analyzed in this cross-sectional trial. The RAPD clinical plus scale was divided into grade 1+, initial weak constriction; grade 2+, initial stall then dilatation; grade 3+, immediate dilatation; and grade 4+, fixed amaurotic pupil. Patients with positive RAPD underwent a visual field examination with Humphrey automated perimetry that included visual field index (VFI), mean deviation (MD), and pattern standard deviation (PSD). Spearman's rank correlation coefficients and linear regression were used to analyze the association between RAPD grades and visual field parameters.

### Results

RAPD clinical plus grades were correlated with interocular VFI (r = 0.55, $P < 0.001$) and MD (r = 0.48, $P = 0.004$) differences. Average interocular VFI differences were estimated as follows: 16.75 × RAPD plus grade– 7.53. RAPD, graded by neutral density filters, was correlated with VFI (r = 0.59, $P < 0.001$), MD (r = 0.54, $P < 0.001$), and PSD (r = 0.34, $P = 0.01$).

### Conclusions

The RAPD plus scale and neutral density filter grading systems were associated with quantitative visual field defect parameters, with VFI showing the strongest association. RAPD clinical grading could substitute more sophisticated central visual field evaluation methods as a low-cost, low-tech, and widely available approach.

**Funding:** JW. was supported by a grant from Faculty of Medicine, Prince of Songkla University, Songkhla, Thailand. The funders had no role in study design, data collection and analysis, decision to publish, or preparation of the manuscript. The funder provided support in the form of salaries for authors [J.W., P.L., and N.A.], but did not have any additional role in the study design, data collection and analysis, decision to publish, or preparation of the manuscript. The specific roles of these authors are articulated in the 'author contributions' section.

**Competing interests:** The authors have declared that no competing interests exist.

## Introduction

Pupillary light reflex is a useful response for anterior visual pathway function assessment [1]. If any pathologic process disrupts the anterior visual function symmetry in one eye, it produces a relative afferent pupillary defect (RAPD) [2]. The neutral density filter bar is a standard RAPD assessment method in which filters of different densities (in log scale units) are placed in front of the better eye before examining the other eye by the swinging light test. A proper-density filter will balance the afferent light source and show no defect. Previous studies have shown that the results of the standard RAPD test are correlated with visual field functions [3–5]. A common, standard method of visual field examination is by Humphrey automated perimetry. This technique often detects visual field loss earlier than does Goldmann kinetic perimetry [1, 6–8]. Moreover, the standard RAPD test requires a neutral density filter, which is not always available in the general clinic, and it is impractical to perform during a routine eye examination. RAPD grading by the clinical plus scale may be a low-cost, low-tech alternative when other assessment methods are not feasible. Previous studies have shown that the two RAPD swinging light test methods, clinical plus grading and neutral density filters, are comparable [9, 10]. Each grade corresponds to a value in log scale units. Therefore, clinicians can use clinical grading instead of the neutral density filters if the latter is not available [9, 10].

To the best of our knowledge, there have been no reports on assessment of the association between RAPD clinical plus scale grades and the visual field parameters. Therefore, we conducted a cross-section clinical trial and aimed to analyze the correlation between RAPD, as graded by the plus scale and neutral density filters, and all visual field parameters, including mean deviation (MD), pattern standard deviation (PSD), and visual field index (VFI). We hypothesized that the clinical RAPD grades evaluated by the swinging light test are associated with changes in the quantitatively assessed visual field parameters. These results could help general ophthalmologists estimate the severity of visual field defects using the clinical plus scale for RAPD in the absence of a specific instrument in routine practice.

## Materials and methods

### Study design

We conducted a prospective cross-sectional study between September 2018 and November 2020 at the Songklanagarind Hospital, a large tertiary care center in southern Thailand. The study protocol was approved by the Human Research Ethics Committee of the Faculty of Medicine, Prince of Songkla University (REC 61-006-2-9) and was conducted according to the tenets of the Declaration of Helsinki. The study was registered with the Thai Clinical Trials Registry (TCTR20180903006). Informed consent was obtained from the participants after explaining to them the nature and possible consequences of the study.

### Sample size

The sample size for this study was calculated by the correlation sample size formula [11]. With 80% power to detect a correlation of 0.4 or greater, assuming a 2-tailed test and a type I error rate of 5%, the required sample size was 47.

### Participants

The study enrolled patients from the eye clinic of Songklanagarind Hospital. Inclusion criteria included age over 20 years, diagnosis of optic neuropathy or retinopathy, and presence of RAPD determined using the swinging light test. Patients were excluded if any of the following criteria were fulfilled: (1) inability to perform a visual field test because of severe visual loss,

such as in patients with RAPD 4+; (2) unreliable visual field test results, defined as >30% false-negative errors or false-positive errors, or >20% fixation loss; (3) presence of anisocoria or pupil abnormality, such as posterior synechiae, iris atrophy, post-laser peripheral iridotomy, or iridoplasty; (4) use of one or more medications affecting the pupil size and reaction, e.g., dilating or constricting eye drops; (5) media opacities that obscured pupillary examination or affected visual field analysis, including cataracts, corneal opacity, and vitreous hemorrhage. The investigator approached all patients who met the inclusion criteria, and those willing to participate in the study provided informed consent.

## Interventions

**RAPD.** RAPD was examined in all participants by two independent investigators (P.L., an ophthalmologic resident, and J.W., a neuro-ophthalmologist) blinded to the visual field defect, using the swinging light test. In case of discrepancy, the RAPD test was conducted, and the results were reviewed by a second neuro-ophthalmologist (N.A.) to reach a consensus. The pupillary examination was performed with the lights in the room turned off in the eye clinic of Songklanagarind Hospital. The observers held a bright, handheld light source (Heine Finoff Transilluminator; HEINE Optotechnik, Herrsching, Germany) approximately 5 cm from the participant's eye, keeping it steady for three seconds while asking the participant to fixate on a distant target (6 m away). After this, the observers swung the light source rapidly to the other eye and held it steady for another three seconds. At this point, the RAPD was recorded on the plus grading scale, as detailed below in accordance with previous reports [1, 7, 9]:

Grade 1+: initial weak constriction, but early redilatation
Grade 2+: no initial constriction (initial stall), then dilatation
Grade 3+: immediate dilatation
Grade 4+: fixed amaurotic pupil or non-reactive pupil in the affected eye

To validate the RAPD light test in all participants, the examiner used a similar light source, with a neutral-density filters bar (Gulden Ophthalmics, Elkins Park, PA, USA) of between 0.3 and 1.8 log scale units, in front of the better eye and gradually increased the filter density from the 0.3 log scale unit. The examiner recorded the density that just balanced the defect in log scale units. If the maximum density (1.8 log scale units) still showed a defect, the examiner recorded this as 1.8 log scale units.

**Visual field examination.** The participants underwent a visual field test with Humphrey automated perimetry program 30–2, following the Swedish Interactive Threshold Algorithms (SITA) standard strategy (Humphrey Field Analyzer; Carl Zeiss Meditec, Dublin, CA, USA), which has good reliability. The test was concluded in approximately 10–30 minutes, depending on the individual. The visual field parameters evaluated by StatPac included MD, PSD, and VFI. Interocular differences in visual field parameters, consisting of interocular MD and VFI differences, were calculated as the visual field parameters of the RAPD-negative eye minus those of the RAPD-positive eye. Interocular PSD difference was calculated as PSD of the RAPD-positive eye minus PSD of the contralateral eye, which yielded a positive value.

## Statistical methods

The data were analyzed by R Core Team version 3.6.1 (2019) [12] and Microsoft Excel, Version 16.16.2 (2018). For descriptive analysis, normally distributed variables are expressed as mean ± standard deviation (SD), and non-normally distributed data are expressed as median and interquartile range. The mixed effect random intercept linear regression model was analyzed to compare visual acuity and visual field parameters between eyes with RAPD and without RAPD. Spearman's rank correlation coefficient was used to evaluate the correlations

between RAPD grades and interocular differences in the visual field parameters. Linear regression analysis was used to assess the relationship between the gold standard RAPD test and clinical plus scale and the relationship between RAPD quantification and the differences between the eyes in visual field outcomes. A difference with $P$-value $< 0.05$ was considered statistically significant.

## Results

### Participant flow

We included 56 patients in the study, recruited from September 2018 to November 2020. All enrolled patients underwent RAPD test using the clinical plus grade and neutral density filter, and a visual field examination with Humphrey automated perimetry. Four patients were excluded because of an unsuccessful visual field test. Finally, we analyzed the test results of 52 patients, as shown in Fig 1.

### Baseline characteristics

All included patients underwent a reliable visual field test. Demographic data and baseline characteristics are summarized in Table 1. The patients' mean age was 48.5 ± 13.0 years. Thirty-two patients (61.5%) had unilateral disease, and 20 (38.5%) had asymmetric bilateral disease. Clinical diagnoses of the eye diseases were classified by etiology into the following six groups: (1) compressive optic neuropathy, 21 (40.4%); (2) anterior ischemic optic neuropathy,

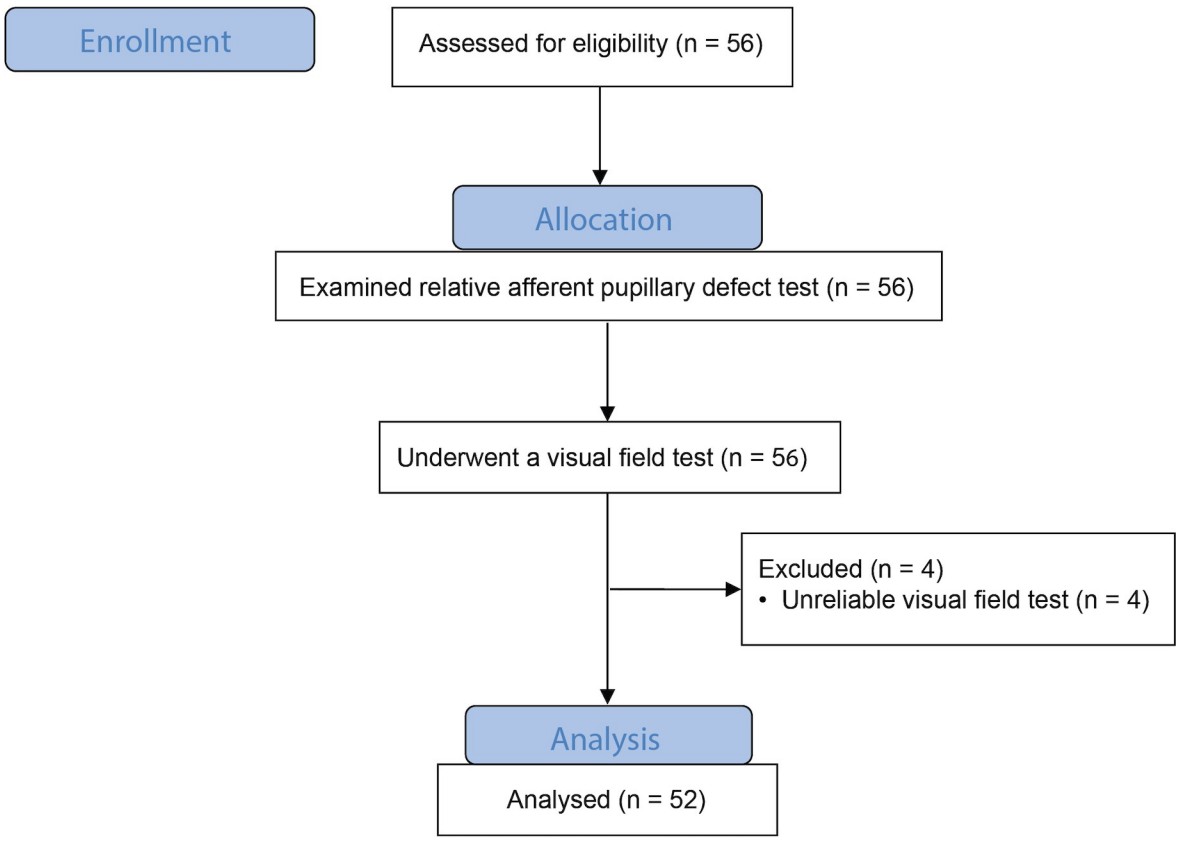

**Fig 1. Participants flow diagram.**

**Table 1. Demographic data and baseline characteristics.**

| Factors | n (%) |
|---|---|
| Gender | |
| Male | 24 (46.2) |
| Female | 28 (53.8) |
| Clinical diagnosis of affected eye | |
| Compressive optic neuropathy | 21 (40.4) |
| Anterior ischemic optic neuropathy | 10 (19.2) |
| Glaucoma | 8 (15.4) |
| Optic neuritis | 7 (13.5) |
| Traumatic optic neuropathy | 2 (3.8) |
| Other causes | 4 (7.7) |
| RAPD by clinical plus scale | |
| 1+ | 10 (19.2) |
| 2+ | 21 (40.4) |
| 3+ | 21 (40.4) |
| RAPD by neutral density filter (log unit) | |
| 0.3 | 2 (3.8) |
| 0.6 | 8 (15.4) |
| 0.9 | 8 (15.4) |
| 1.2 | 8 (15.4) |
| 1.5 | 5 (9.6) |
| 1.8 | 21 (40.4) |

RAPD, relative afferent pupillary defect.

10 (19.2%); (3) glaucoma, eight (15.4%); (4) optic neuritis, seven (13.5%); (5) traumatic optic neuropathy, two (3.8%); and (6) other causes, including neuroretinitis, papillophlebitis with a secondary branch retinal vein occlusion, optic atrophy supposedly from an old central retinal artery occlusion, and postvitrectomized eye from rhegmatogenous retinal detachment and secondary ocular hypertension, four (8.51%) patients. Two examiners performed RAPD test using the clinical plus scale and neutral density filter as the gold standard for all patients. The majority of the results were similar between both observers, and for two of the 52 patients (3.8%), evaluation by the third examiner was necessary. All patients with RAPD were categorized into three groups using the clinical plus scale and six groups using the neutral density filter (Table 1). Groups 3+ and 2+ included 21 patients (40.4%) each. A high rate of the 1.8 log scale neutral density filter was noted in our study (40.4%). The mean logMAR visual acuity, MD, PSD, and VFI in the RAPD-positive eyes are summarized in Table 2. There were significant differences between RAPD-positive eyes and contralateral eyes for all visual field parameters.

## Correlation of RAPD grading by the plus scale and neutral density filters

The RAPD plus scale grades were strongly positively correlated with the neutral density filters used (r = 0.94, $P < 0.001$). The linear regression model indicated $R^2 = 0.88$ ($P < 0.001$).

## Correlation between RAPD by the plus scale and visual field parameters

Spearman's rank correlations between RAPD plus grades and the three visual field parameters are presented in Table 3. A moderately positive correlation was noted between the RAPD plus

**Table 2. Visual acuity and visual field parameters between RAPD-positive eyes and contralateral eyes.**

| Parameters | RAPD-positive eyes (n = 52) | | Contralateral eyes (n = 52) | | P-value |
| --- | --- | --- | --- | --- | --- |
| | Mean ± SD | Mean (95% CI)* | Mean ± SD | Mean (95% CI)* | |
| Visual acuity (logMAR) | 0.31 ± 0.30 | 0.31 (0.25, 0.38) | 0.12 ± 0.13 | 0.12 (0.06, 0.18) | <0.001 |
| Mean deviation (dB) | −13.10 ± 7.73 | −13.10 (−14.75, −11.46) | −2.95 ± 3.86 | −2.95 (−4.60, −1.31) | <0.001 |
| Pattern standard deviation (dB) | 9.97 ± 4.53 | 9.97 (8.80, 11.14) | 3.90 ± 4.12 | 3.90 (2.73, 5.06) | <0.001 |
| Visual field index (%) | 64.38 ± 24.22 | 64.38 (59.35, 69.42) | 93.90 ± 10.68 | 93.90 (88.86, 98.94) | <0.001 |

SD, standard deviation; CI, confidence interval; logMAR, Logarithm of the Minimum Angle of Resolution; dB, decibel; RAPD, relative afferent pupillary defect.

*Derived from mixed effect random intercept linear regression model

grades and interocular MD differences and interocular VFI differences. The correlation between the RAPD plus grades and PSD differences was insignificant, as shown in Table 3.

Linear regression was calculated for the estimated values of interocular differences in the visual field parameters for each RAPD plus grade. The RAPD plus grades were positively predictive of MD and VFI differences, as shown in Fig 2A and 2B. On average, for each RAPD plus grade increase, the MD differences increased by 5 dB, and the following formula could be used to estimate the VFI difference: interocular VFI difference = 16.75 × RAPD grade– 7.53.

## Correlation between RAPD neutral density filter grades and visual field parameters

All correlations between RAPD neutral density filter grades and interocular MD, PSD, and VFI differences were significant, as shown in Table 2. The strongest correlation coefficient was found between RAPD neutral density filter grades and VFI differences (r = 0.59, $P < 0.001$). From the regression line in Fig 2C, it was estimated that for every 0.3 log RAPD grade unit increase, there was an MD difference increase of 2.3 dB. The estimated VFI differences were calculated using the following formula, based on Fig 2D: interocular VFI difference = 27.37 × RAPD in log units– 6.

## Discussion

Clinical assessment of the pupillary light reflex is useful for the evaluation of disrupted visual function in many pathologic processes during a routine examination. Although evaluation of the visual field using Humphrey automated perimetry plays a fundamental role in diagnosing visual pathway disorders and the swinging light test with neutral density filters can be used to assess the severity of visual function abnormalities, the facilities for these tests are not always available at clinics. In search of an alternative, widely available approach, we designed this

**Table 3. Spearman's rank correlation coefficient between interocular differences in visual field parameters and RAPD grades by the clinical plus scale and neutral density filter.**

| Visual field parameters | RAPD by plus grade | | RAPD by neutral density filter | |
| --- | --- | --- | --- | --- |
| | r (95% CI) | P-value | r (95% CI) | P-value |
| MD | 0.48 (0.26–0.70) | 0.004 | 0.54 (0.33–0.74) | < 0.001 |
| PSD | 0.25 (−0.04–0.54) | 0.07 | 0.34 (0.06–0.62) | 0.01 |
| VFI | 0.55 (0.34–0.76) | < 0.001 | 0.59 (0.39–0.79) | < 0.001 |

MD, mean deviation; PSD, pattern standard deviation; VFI, visual field index; RAPD, relative afferent pupillary defect; CI, confidence interval.

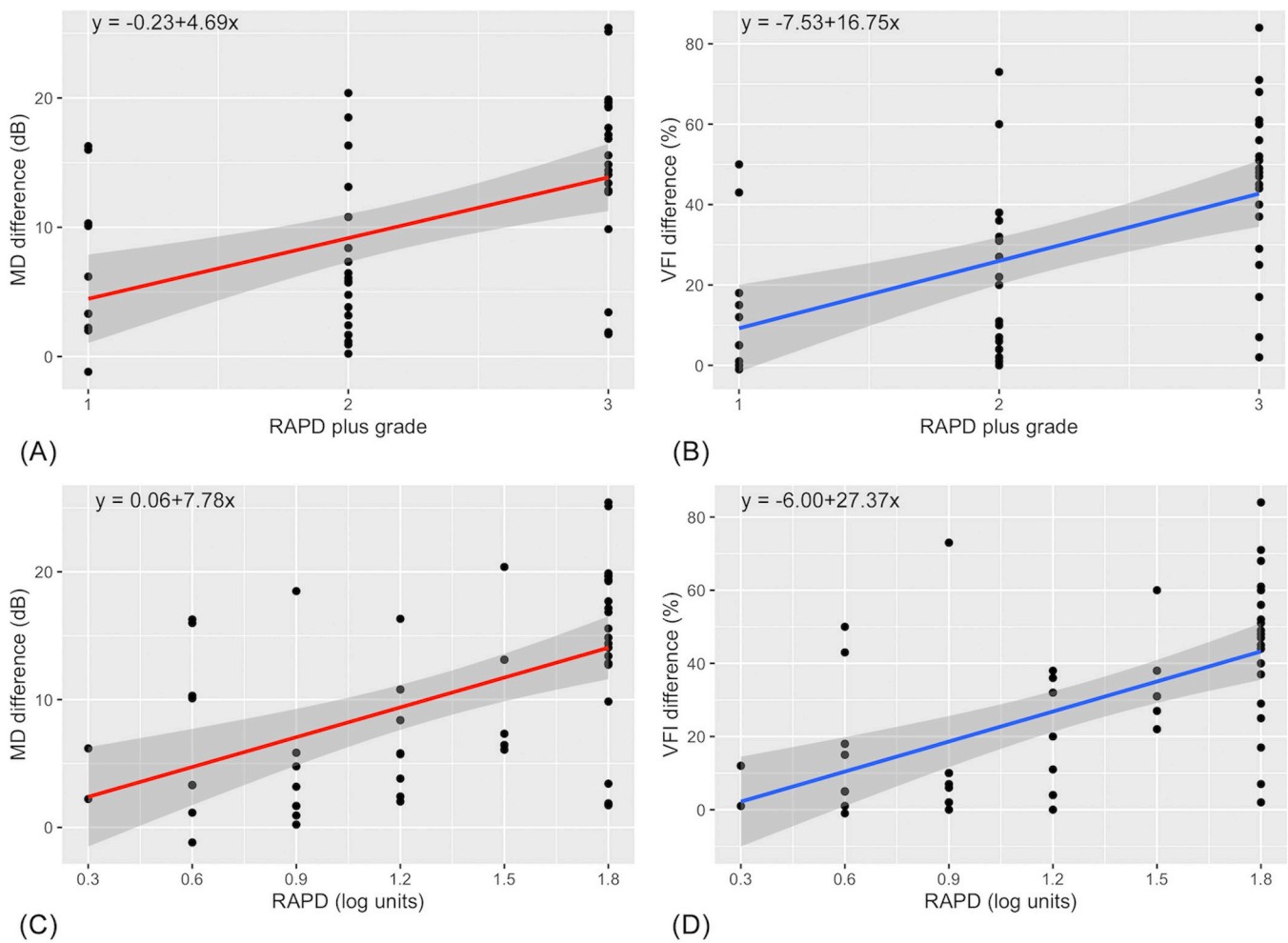

**Fig 2. Linear regression analysis. (A)** Linear regression of relative afferent pupillary defect (RAPD) plus grades and interocular mean deviation (MD) differences ($R^2 = 0.24$, $P < 0.001$); **(B)** linear regression of RAPD plus grades and interocular visual field index (VFI) differences ($R^2 = 0.28$, $P < 0.001$); **(C)** linear regression of RAPD neutral density filter grades and interocular MD differences ($R^2 = 0.29$, $P < 0.001$); **(D)** linear regression of RAPD neutral density filter grades and interocular VFI differences ($R^2 = 0.33$, $P < 0.001$).

prospective clinical trial to assess the correlation between RAPD, graded by the clinical plus scale and neutral density filters, and central visual field parameters, assessed by Humphrey automated perimetry. To our knowledge, the association between clinical RAPD grades and changes in quantitatively assessed visual field parameters has been demonstrated for the first time. The correlation between RAPD and VFI was stronger than that with other visual field parameters. We also found a strong association between the clinical plus scale and neutral density filters in their assessment of the RAPD.

Clinical identification of RAPD using the swinging light test is sensitive for detecting optic nerve disorders or substantial retinal disease. In our study, we assessed the quantification of RAPD in patients who had various etiologies of optic neuropathy or retinopathy, such as retinal artery occlusion and retinal detachment. Regarding the baseline characteristics, most patients had severe RAPD by using the clinical plus scale (grade 2+ to 3+) and neutral density filters (1.8 log unit). The average Humphrey VFI of the RAPD-positive eyes was 64.4%, indicating profound visual field loss, while the mean logMAR visual acuity was 0.3, considered

nearly normal vision. Therefore, RAPD may be present in patients with normal visual acuity who had suffered significant visual field defects.

Previous studies have shown a positive correlation between the results of the standard RAPD test and interocular MD differences in the computed tomography visual field (CTVF) 30–2 pattern, in agreement with our results [4, 5, 8]. Johnson et al. found a correlation of r = 0.69 between RAPD neutral density filter grades and MD differences in 26 patients with optic nerve disease [4]. Kardon et al. evaluated RAPD neutral density filter grades and MD differences in 137 patients (84 positive and 53 negative for RAPD), observing a correlation of r = 0.66 [5]. The study by Lagrèze and Kardon reported a strong positive relationship between RAPD with MD of the affected eyes with optic neuropathy ($r^2$ = 0.52) [8]. Recently, Schiefer et al. found the correlations between standard RAPD and MD differences and loss volume to be higher than 20˚ and 10˚, respectively, when assessed by the visual field test within eccentricities of 30˚. Their assessment was based on an individually modeled 3D hill of vision on OCTOPUS 101 Perimeter Visual Field Analyzer (Haag-Streit Inc., Koeniz, Switzerland) [13]. Following that study, we elected to use a visual field of 30˚, rather than visual fields of 20˚ and 10˚, to better cover the retinal nerve fiber layers and ganglion cell defects. Additionally, recent research assessed the correlation between RAPD scores measured by an automated pupillometer and through interocular MD differences, demonstrating the significantly moderate relationship ($r^2$ = 0.36) in patients with optic neuropathy [14]. Similar results were also reported in patients with glaucoma, assessed using the CTVF 24–2 pattern and automated pupillometer [15–18].

Interestingly, our study revealed the correlation of RAPD with interocular VFI differences to be stronger than that with MD differences. VFI is a visual field parameter weighing the percentage of loss by central visual field involvement. The percentage of VFI when the visual field defect affects the central vision will be lower than that with a peripheral visual defect [19]. The retinal nerve fiber layers are more densely packed on the temporal side of the optic disc than on the nasal side. This affects functional changes in the central visual field if the retinal nerve fiber layers around the disc are damaged. It might also cause changes in the disc and RAPD [20]. Several studies have suggested that structural changes to the retinal nerve fiber and ganglion cell layers affect RAPD, with a strong correlation [8, 15, 17]. Therefore, we assumed that the VFI, more than MD, would correlate with the retinal nerve fiber layers that affect RAPD. A recent study reported a significant negative correlation between RAPD, assessed by pupillometer, and the VFI, on the basis of CTVF 24–2 (r = −0.68). This correlation coefficient was slightly lower than that between RAPD and MD (r = −0.73) in glaucomatous optic neuropathy [17]. This contrasted with our results because the peripheral vision of patients with glaucomatous optic neuropathy in CTVF 24–2 was more impaired than their central vision. Glaucoma patients with severe RAPD had good VFI despite their poor MD.

Our study also revealed that RAPD, measured by clinical plus grades, was not correlated with PSD. This result was because PSD is a mean value of the difference in the hill of vision height and does not represent the nerve fiber layer function. Hence, its value might be low in patients with normal visual field or generalized poor central vision. Additionally, it might be high in cases of deep and focal visual loss [21]. On the basis of these concepts, we could anticipate that PSD would not correlate strongly with RAPD.

Limitations to this study could result from the diversity of clinical diagnoses among patients with abnormal RAPD. Furthermore, some optic nerve diseases, such as optic neuritis, were unstable and possibly unaffected by the visual field during the recovery periods. Even though patients with optic neuritis had recovery of visual field defects, their RAPD could remain positive for up to two years [22]. Thus, it is important to consider the clinical correlation individually with a large sample size in the future. Besides, assessment of the RAPD plus scale required

clinical experience to recognize the pupil reaction and differentiate among the grades of RAPD test. In our study, two patients were graded differently by an ophthalmologic resident and a neuro-ophthalmologist. We recommend video recording the RAPD test to re-evaluate the plus scale grading if the result is questionable and to assess the variability in the result obtained when the test is performed by experienced ophthalmologists and general practitioners in future research. These results would enable standardization of the test among examiners and could be useful for the application of the clinical RAPD test in routine general practice. Additionally, the presence of RAPD can correlate with damage to the retinal nerve fiber layers in the presence of unilateral optic nerve disease but cannot justify the importance of the visual field in relation to visual pathway disorders.

## Conclusions

We found that the RAPD grades, evaluated by the clinical plus scale and neutral density filters, were correlated with changes in the quantitative visual field parameters assessed by Humphrey automated perimetry, with the VFI showing the strongest correlation. These findings suggest that RAPD clinical plus grading could substitute more sophisticated central visual field evaluation methods when other assessment methods are unavailable.

## Supporting information

**S1 Checklist.**
(PDF)

**S1 File. Raw data.**
(CSV)

## Acknowledgments

The authors acknowledge with appreciation Prof. Edward McNeil and Dr. Alan Geater, from the Faculty of Medicine, Prince of Songkla University, for their assistance with the statistical analysis. We also would like to thank Editage (www.editage.com) for English language editing.

## Author Contributions

**Conceptualization:** Juthamat Witthayaweerasak.

**Data curation:** Juthamat Witthayaweerasak, Pemika Lertjittham, Nipat Aui-aree.

**Formal analysis:** Juthamat Witthayaweerasak, Pemika Lertjittham.

**Funding acquisition:** Juthamat Witthayaweerasak.

**Investigation:** Nipat Aui-aree.

**Methodology:** Juthamat Witthayaweerasak, Pemika Lertjittham, Nipat Aui-aree.

**Project administration:** Nipat Aui-aree.

**Resources:** Juthamat Witthayaweerasak.

**Software:** Juthamat Witthayaweerasak.

**Supervision:** Juthamat Witthayaweerasak, Nipat Aui-aree.

**Validation:** Juthamat Witthayaweerasak, Pemika Lertjittham.

**Visualization:** Pemika Lertjittham, Nipat Aui-aree.

**Writing – original draft:** Juthamat Witthayaweerasak, Pemika Lertjittham.

**Writing – review & editing:** Juthamat Witthayaweerasak.

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
