## [Decision Letter · Decision Letter 0]

7 Mar 2022

PONE-D-21-21722Correlation between relative afferent pupillary defect and visual field defects on Humphrey automated perimetry: A cross-sectional clinical trialPLOS ONE

Dear Dr. Witthayaweerasak,

Thank you for submitting your manuscript to PLOS ONE. After careful consideration, we feel that it has merit but does not fully meet PLOS ONE’s publication criteria as it currently stands. Therefore, we invite you to submit a revised version of the manuscript that addresses the points raised during the review process. Please submit your revised manuscript by Apr 21 2022 11:59PM. If you will need more time than this to complete your revisions, please reply to this message or contact the journal office at plosone@plos.org. Please include the following items when submitting your revised manuscript:A rebuttal letter that responds to each point raised by the academic editor and reviewer(s). You should upload this letter as a separate file labeled 'Response to Reviewers'.A marked-up copy of your manuscript that highlights changes made to the original version. You should upload this as a separate file labeled 'Revised Manuscript with Track Changes'.An unmarked version of your revised paper without tracked changes. You should upload this as a separate file labeled 'Manuscript'.

We look forward to receiving your revised manuscript.

Kind regards,

Oana Dumitrascu, M.D.

Academic Editor

PLOS ONE

Journal Requirements:

2. Thank you for stating the following financial disclosure: "JW. was supported by a grant from Faculty of Medicine, Prince of Songkla University, Songkhla, Thailand. The funders had no role in study design, data collection and analysis, decision to publish, or preparation of the manuscript."

We note that one or more of the authors is affiliated with the funding organization, indicating the funder may have had some role in the design, data collection, analysis or preparation of your manuscript for publication; in other words, the funder played an indirect role through the participation of the co-authors. If the funding organization did not play a role in the study design, data collection and analysis, decision to publish, or preparation of the manuscript and only provided financial support in the form of authors' salaries and/or research materials, please do the following:

a. Review your statements relating to the author contributions, and ensure you have specifically and accurately indicated the role(s) that these authors had in your study. These amendments should be made in the online form.

b. Confirm in your cover letter that you agree with the following statement, and we will change the online submission form on your behalf: 

“The funder provided support in the form of salaries for authors [insert relevant initials], but did not have any additional role in the study design, data collection and analysis, decision to publish, or preparation of the manuscript. The specific roles of these authors are articulated in the ‘author contributions’ section.

Additional Editor Comments (if provided):

Add the n to table 1: eyes with and without RAPD. Mention how many patients had RAPD and their grades by clinical plus criteria, and how many patients had RAPD by neutral density filters, and their mean grading.

Reviewers' comments:

Reviewer's Responses to Questions

**Comments to the Author**

1. Is the manuscript technically sound, and do the data support the conclusions?

Reviewer #1: Yes

Reviewer #2: Yes

2. Has the statistical analysis been performed appropriately and rigorously? 

Reviewer #1: Yes

Reviewer #2: I Don't Know

3. Have the authors made all data underlying the findings in their manuscript fully available?

Reviewer #1: No

Reviewer #2: Yes

4. Is the manuscript presented in an intelligible fashion and written in standard English?

Reviewer #1: Yes

Reviewer #2: Yes

5. Review Comments to the Author

Reviewer #1: This is a straightforward and useful trial. However, the authors did not include much information about the enrolled patient demographics. Please provide a table with the subject characteristics (age, sex, diagnosis, monocular vs binocular disease, visual acuity, etc.) and details on how these related to their study-related outcomes of RAPD and HVF metrics.

Reviewer #2: How you may justify the absence of RAPD due to symmetric optic nerve damage?

The presence of RAPD can correlate with the damage to RNFL in the presence of unilateral optic nerve disease but can not justify the importance of visual field in relation to visual pathway disorders. It will be good to mention that in the paper. Also, visual field plays a fundamental role for diagnosis of visual pathway disorder.

Other Editorial Comments:

Add the n to table 1: eyes with and without RAPD. Mention how many patients had RAPD and their grades by clinical plus criteria, and how many patients had RAPD by neutral density filters, and their mean grading.

6. PLOS authors have the option to publish the peer review history of their article (what does this mean?). If published, this will include your full peer review and any attached files.

Reviewer #1: No

Reviewer #2: **Yes: **Nafiseh Hashemi, MD

---

## [Author Response · Author response to Decision Letter 0]

27 Mar 2022

We sincerely appreciate the efforts of the editor and reviewers in the thorough review of our manuscript. We have addressed all the journal requirements and the valuable comments of the editor and reviewers in a separate file labeled "Response to Reviewers".

---

## [Decision Letter · Decision Letter 1]

11 Apr 2022

Correlation between relative afferent pupillary defect and visual field defects on Humphrey automated perimetry: A cross-sectional clinical trial

PONE-D-21-21722R1

Dear Dr. Witthayaweerasak,

We’re pleased to inform you that your manuscript has been judged scientifically suitable for publication and will be formally accepted for publication once it meets all outstanding technical requirements.

Kind regards,

Oana Dumitrascu, M.D.

Academic Editor

PLOS ONE

Additional Editor Comments (optional):

Reviewers' comments:

Reviewer's Responses to Questions

**Comments to the Author**

1. If the authors have adequately addressed your comments raised in a previous round of review and you feel that this manuscript is now acceptable for publication, you may indicate that here to bypass the “Comments to the Author” section, enter your conflict of interest statement in the “Confidential to Editor” section, and submit your "Accept" recommendation.

Reviewer #1: All comments have been addressed

2. Is the manuscript technically sound, and do the data support the conclusions?

Reviewer #1: Yes

3. Has the statistical analysis been performed appropriately and rigorously? 

Reviewer #1: Yes

4. Have the authors made all data underlying the findings in their manuscript fully available?

Reviewer #1: Yes

5. Is the manuscript presented in an intelligible fashion and written in standard English?

Reviewer #1: Yes

6. Review Comments to the Author

Reviewer #1: The revisions addressed my previous comments.

The statement on line 298-301 does not make sense, particularly the part about "cannot justify the importance of

the visual field in relation to visual pathway disorders." Please correct or omit this statement.

7. PLOS authors have the option to publish the peer review history of their article (what does this mean?). If published, this will include your full peer review and any attached files.

Reviewer #1: No

---

## [Editor Report · Acceptance letter]

18 May 2022

PONE-D-21-21722R1 

Correlation between relative afferent pupillary defect and visual field defects on Humphrey automated perimetry: A cross-sectional clinical trial 

Dear Dr. Witthayaweerasak:

I'm pleased to inform you that your manuscript has been deemed suitable for publication in PLOS ONE. Congratulations! Your manuscript is now with our production department. 

Kind regards, 

on behalf of

Dr. Oana Dumitrascu 

Academic Editor

PLOS ONE